# Risk of Subsequent Primary Cancer in Thyroid Cancer Survivors: A Nationwide Population-Based Study

**DOI:** 10.3390/diagnostics13182903

**Published:** 2023-09-11

**Authors:** Min-Su Kim, Sang Jun Lee, Myeong Hoon Lee, Jay Hyug Choi, Hyun Wook Han, Young Shin Song

**Affiliations:** 1Department of Otorhinolaryngology-Head and Neck Surgery, CHA Bundang Medical Center, CHA University School of Medicine, CHA University, Seongnam 13496, Republic of Korea; 2Department of Biomedical Informatics, CHA University School of Medicine, CHA University, Seongnam 13496, Republic of Korea; 3Department of Internal Medicine, Seoul Metropolitan Government Seoul National University Boramae Medical Center, Seoul National University College of Medicine, Seoul 07061, Republic of Korea

**Keywords:** thyroid neoplasms, second primary neoplasms, risks, adult

## Abstract

**Background:** Despite obtaining a good prognosis and long life expectancy, survivors of thyroid cancer can nevertheless develop subsequent primary cancer (SPC). We investigated the risk and types of SPC in patients with thyroid cancer and compared them with subjects without thyroid cancer history (controls). **Methods:** We conducted a nationwide, population-based, retrospective cohort study based on the Korean National Health Insurance Database. A total of 432,654 patients diagnosed with thyroid cancer between 2004 and 2019 were 1:1 matched with controls for age, sex, income, and region of residence. The hazard ratios (HR) and 95% confidence intervals (CI) of SPC were estimated using Cox proportional hazard models. **Results:** In total, 78,584 (18.2%) patients with thyroid cancer and 49,979 (11.6%) controls were diagnosed with SPCs over a mean follow-up of 6.9 years. Patients with thyroid cancer had a higher risk of SPC at any site (adjusted HR, 1.62; 95% CI, 1.60–1.64) than the controls. The risk of SPCs was particularly high for patients diagnosed with thyroid cancer at a younger age (<40 years) and within 5 years. **Conclusions:** Medical caregivers should consider the long-term follow-up of patients with thyroid cancer and discuss the risk of SPC, especially if they complain of cancer-related symptoms.

## 1. Introduction

Thyroid cancer is the most common endocrine malignancy worldwide and has the highest prevalence among all cancers in Korea. In 2019, thyroid cancer accounted for 21.5% of all cancers, and the age-standardized prevalence rate was 563.7 per 100,000 individuals. Among cancers, it has the largest number of long-term survivors, with a 5-year relative survival rate of 100.0% [1]. Moreover, thyroid cancer affects relatively young individuals, especially women aged <40 years. Considering a long life expectancy, the occurrence of subsequent primary cancer (SPC) among survivors is an important concern for both patients and caregivers [2].

Previous cohort studies have reported an increased risk of SPC in thyroid cancer survivors [3,4,5,6]. A European cohort study found an overall increased risk for SPC of 27%. Particularly, radioactive iodine (RAI) administration was associated with the occurrence of bone and soft tissue, colorectal, and salivary gland cancers. However, the study evaluated a relatively small number of patients in a short follow-up period of 2 years [3]. Three studies by the U.S. Surveillance, Epidemiology, and End-Results (SEER) Program reported that the risk of SPC after thyroid cancer was higher than that expected for the general population [4,5,6]. Patients treated with RAI and younger patients had a greater risk of SPC. Furthermore, the greatest risk of SPC occurred within 5 years of thyroid cancer diagnosis [5].

A large Korean population study that included 178,844 registrants with thyroid cancer from the Korea Central Cancer Registry Database between 1993 and 2010 demonstrated that the risks of several cancers were significantly increased during follow-up [2]. However, the treatment of thyroid cancer has evolved significantly since, and we are now in an era of fewer surgeries and RAI therapy for thyroid cancer. After 2012, although the incidence of thyroid cancer and total thyroidectomies declined in Korea, the rate of partial thyroidectomies gradually increased, and, accordingly, the rate of RAI decreased [7]. Furthermore, changes have been made to the Korean database to include more information, such as socioeconomic state, region, and comorbidities, than was analyzed in the previous Korean study. Therefore, this updated Korean national population-based study evaluated the risk of SPC in patients with thyroid cancer and compared it with subjects without a history of thyroid cancer in a larger population than previously studied.

## 2. Materials and Methods

### 2.1. Data Source

The Korean National Health Insurance Service (NHIS) was established under the Ministry of Health and Welfare to provide mandatory health insurance services to about 97% of the Korean population [8]. This study used a customized database derived from the nationwide Korean NHIS. A customized database refers to data that are processed and provided as demand-tailored data so that health information data collected, held, and managed by the NHIS can be used for policy and academic research purposes. The database included information on medical use behaviors, diagnostic codes, prescription codes, and drug codes of all citizens.

### 2.2. Study Population

From the Korean population registered in the NHIS between January 2002 and December 2019, individuals with a history of thyroid cancer (International Classification of Diseases 10th Revision [ICD-10], C73; *n* = 776,413) were extracted as cases. Initially, individuals without a record of thyroid cancer who were 1:2 matched in age, sex, and region of residence to cases were selected as controls (*n* = 1,473,314). To ensure the inclusion of patients newly diagnosed with thyroid cancer, we excluded patients who had not undergone thyroidectomy (claim codes: P4551, P4552, P4553, P4554, and P4561) and those diagnosed with thyroid cancer from 2002 to 2003. The initial 2 years (2002 and 2003) were excluded as a washout period due to the possibility of including patients diagnosed with thyroid cancer prior to the study period. From the participants (with and without thyroid cancer), we excluded individuals diagnosed with antecedent malignancies within 2 years of thyroid cancer diagnosis, as well as individuals with missing information on age, sex, region of residence, and income level. The final number of patients with thyroid cancer was 432,654. Patients with thyroid cancer were 1:1 matched with participants without thyroid cancer using propensity score matching for age, sex, region of residence, and income level. The index dates for the participants with and without thyroid cancer were set as the time of diagnosis of thyroid cancer and the index date of the matched participants, respectively. Finally, 432,654 thyroid cancer cases and 432,654 matched controls were included (Figure 1).

### 2.3. Outcomes (Subsequent Primary Cancers) and Covariates

SPCs were defined based on ICD-10 codes (C00–C26, C30–C58, and C60–C97), which occurred after the index date, except for thyroid and metastatic cancers (C73 and C77–C80). The following covariates were used for multivariate analyses: age (continuous; years), sex (categorical; male or female), region of residence (categorical; urban or rural), income level (categorical; 0–4, 5–8, 9–12, 13–16, or 17–20), body mass index (BMI; continuous; kg/m^2^), smoking status (categorical; never, previous, or current), and Charlson comorbidity index (CCI; categorical; 0, 1, 2, 3, or ≥4). CCI was calculated as described in a previous study [9].

### 2.4. Statistical Analysis

Continuous and categorical variables were presented as mean ± standard deviation and number (%), respectively. The *t*-test was used for continuous variable analyses, whereas the chi-square test was used for categorical variable analyses. The Cox proportional hazards models were adopted to evaluate the crude hazard ratio (HR) and adjusted HR (aHR) for CCI with 95% confidence intervals (CI). For the subgroup analyses, the participants were divided by age (<40 and ≥40 years), sex (male and female), and latency period (<2.0, 2.0–4.9, 5.0–9.9, and ≥10.0 years), the latter of which is the duration from the index date to the date of diagnosis of SPC. Cox regression analyses were performed for each subgroup. Because obesity and smoking are well-established risk factors for several cancers, further subgroup analysis of participants with information on BMI and smoking status from the NHIS Health Screening Program data was conducted to reduce confounding bias. The Cox models were adjusted for age, sex, region of residence, income level, CCI, obesity, and smoking status. Statistical significance was defined as a two-sided *p*-value < 0.05. R version 4.1.0 (R Foundation for Statistical Computing, Vienna, Austria) was used to perform all statistical analyses.

## 3. Results

The participant characteristics are shown in Table 1. The mean age was 48.6 ± 12.2 years, and 80.9% of the study participants were female. No significant differences were found in age, sex, income, and region of residence between the two groups after propensity score matching.

During the mean follow-up of 6.9 years, 78,584 (27.9 per 1000 person years) and 49,979 (15.2 per 1000 person years) SPCs occurred in the thyroid cancer and control groups, respectively (Table 2). Overall, the risk of SPC of all sites was elevated in patients with thyroid cancer (aHR, 1.62; 95% CI, 1.60–1.64; Table 2). The results are consistent in the subgroup with known BMI and smoking information, even after additional adjustment for obesity and smoking status (aHR, 1.58; 95% CI, 1.55–1.62; Appendix A). The SPCs with the greatest risks of occurrence were bone and soft tissue, breast, and adrenal gland cancers in male patients with thyroid cancer (aHR, 4.92, 10.48, and 6.58; 95% CI, 3.15–7.69, 4.78–22.95, and 3.09–14.00, respectively); whereas in female patients, the SPCs were bone and soft tissue cancer, adrenal cancer, and leukemia (aHR, 2.55, 3.71, and 2.67; 95% CI, 2.04–3.17, 2.78–4.96, and 2.22–3.21, respectively; Figure 2A,B; Appendix A). The risk for SPC was especially high in patients with thyroid cancer under the age of 40 years (aHR, 1.95; 95% CI, 1.90–2.01). The risks of bone and soft tissue cancer and adrenal cancer were highest in both the thyroid cancer group aged < 40 years (aHR, 4.38 and 6.66; 95% CI, 2.54–7.55 and 3.38–13.12, respectively) and ≥40 years (aHR, 2.80 and 3.63; 95% CI, 2.27–3.45 and 2.71–4.88, respectively; Figure 2C,D; Appendix A).

In analyzing the risks of SPC according to the latency period, we found that the risks remarkably increased in latency periods of <2 and 2.0–4.9 years (aHR, 3.12 and 2.21; 95% CI, 3.05–3.18 and 2.16–2.26, respectively; Figure 3A,B; Appendix A). However, after 5 years of thyroid cancer diagnosis, patients showed only a modest increase in the risk of SPCs (aHR, 1.12 and 1.21; 95% CI, 1.10–1.15 and 1.16–1.25 for latency periods of 5.0–9.9 and ≥10 years, respectively; Figure 3C,D; Appendix A). Notably, the SPCs with exceptionally high risk for each latency period were esophagus cancer for <2 years (aHR, 9.24; 95% CI, 5.88–14.52) and leukemia for 2.0–4.9 years (aHR, 6.61; 95% CI, 4.59–9.53). In addition, the risks of developing bone and soft cancer remained especially high (aHR, 5.76 and 3.64; 95% CI, 5.88–14.52 and 2.43–5.46, for latency periods of <2 and 2.0–4.9 years, respectively) and adrenal cancer (aHR, 6.12, 8.53, and 3.54; 95% CI, 3.96–9.46, 4.65–15.64, and 2.05–6.13, for latency periods of <2, 2.0–4.9, and 5.0–9.9 years, respectively) for latency periods of <5 and <10 years, respectively (Figure 3).

## 4. Discussion

In the present study, we found that patients with thyroid cancer were associated with an increased risk of SPCs, particularly for bone and adrenal gland cancers, compared with matching controls without thyroid cancer. Younger patients had a higher risk of SPC, and patients diagnosed with thyroid cancer within 5 years have an especially high risk of SPC. In addition, the risks of esophageal cancer within 2 years of thyroid cancer diagnosis and of leukemia between 2 and 5 years after thyroid cancer diagnosis were high.

Several explanations can be made regarding the elevated risk for SPCs in younger survivors. Although younger patients with thyroid cancer carry a greater risk of extensive disease at clinical presentation, such as lymph node metastasis, extrathyroidal extension, and pulmonary metastasis, they have a better prognosis and longer survival times than older patients. Younger patients are also more likely to undergo more aggressive treatment, such as total thyroidectomy and adjuvant RAI therapy, given the initial clinical presentation. Several studies have reported an elevated risk of long-term SPCs from the use of RAI therapy in patients with thyroid cancer [10,11,12,13]. Moreover, the higher risk of SPCs in young thyroid cancer survivors in our study is consistent with the findings of previous studies [10,12].

In the present study, the risk of SPC was markedly higher within 5 years of thyroid cancer diagnosis, especially within the first 2 years. The U.S. SEER database study showed that the standardized incidence ratio (SIR), which means the incidence rate in the study cohort compared with that in the general population, was greatly increased immediately after diagnosis and decreased rapidly by 5 years after diagnosis [5]. Moreover, a previous Korean study reported that 66% and 25% of SPCs occurred within <5 and 5–10 years of thyroid cancer diagnosis, respectively, during the follow-up period [2]. This can be explained by surveillance bias during or after thyroid cancer diagnosis. However, in our study, SPCs with the most significantly elevated risk within the first 5 years included radiogenic malignancies, such as leukemia, and genetically involved cancers, such as adrenal gland cancer. Thus, we cannot exclude the possibility of a therapeutic effect and an environmental or genetic mechanism [2,14]. Moreover, the screening and follow-up tests for most thyroid cancer cases are limited to neck ultrasonography and blood tests, including thyroid-specific markers. We also matched participants by income, which indirectly reflects accessibility to hospitals.

A U.S. population-based cancer registry dataset derived from the SEER program showed that bone sarcomas were rare (*n* = 50), but the risk (SIR, 2.7; 95% CI, 1.0–5.8) among young adult thyroid cancer survivors over the general population was statistically significantly elevated [15]. Unsurprisingly, bone sarcoma is related to the increased cumulative administration of RAI or radiation [16]. Younger thyroid cancer survivors are exposed to more frequent diagnostic medical radiation from neck and chest CT or PET-CT, which are independent risk factors for the development of SPCs in patients with thyroid cancer. Therefore, physicians should minimize exposure to high-radiation doses from radiologic examinations of young patients with treated thyroid cancer [17].

A strong relationship with thyroid cancer and subsequent adrenal gland cancer can be attributed to multiple endocrine neoplasia type 2 (MEN2). MEN2 is a hereditary cancer syndrome caused by *RET* proto-oncogene mutations involving multiple endocrine organs, including the thyroid and adrenal glands [18]. Medullary thyroid cancer (MTC) is usually the first and most frequent manifestation of MEN2 with 100% penetrance. Pheochromocytoma, a neuroendocrine tumor that typically develops in the adrenal medulla, occurs in 40–50% of patients with MEN2. MTC accounts for 3–5% of thyroid cancer cases. Close to 25% of cases are familial, while 75% are considered sporadic. Germline *RET* mutations are associated with autosomal-dominant inherited MEN2 [19]. In MTC, preoperative genetic counseling is crucial in the initial diagnosis, and investigating the possibility of MEN2 is necessary for the diagnosis and follow-up.

This study has several advantages. To our knowledge, this is the largest study that evaluated the risk of SPC in patients with thyroid cancer and compared it with subjects without thyroid cancer. In addition, the results of our study are representative of the entire Korean population. Another strength is the availability of comprehensive medical records for each participant without being affected by recall bias since patients’ medical records were extracted from the NHIS database. Finally, we matched patients with thyroid cancer with controls by age, sex, income, and region of residence to control for major confounding factors and even confirmed the consistency of results after further adjustment for confounders, such as the CCI score, BMI, and smoking status. Moreover, the risk assessment of SPC was conducted in various subgroups.

Nevertheless, our study had some limitations. First, although we confirmed the association between thyroid cancer and SPCs, proving the causality was difficult because multiple factors are involved in carcinogenesis. Thus, we tried to control the confounding factors using propensity score matching, multivariate-adjusted models, and subgroup analyses. However, the effect of treatment, such as the dose of RAI and external beam radiation, was not considered, which is a limitation. Further studies on the causal relationship and mechanisms are needed. Second, due to limitations of the NHIS database, the cancer stages and histologic subtypes of thyroid cancer, such as papillary, follicular, medullary, and anaplastic subtypes, were not available.

## 5. Conclusions

This largest population-based national study showed that the risks of SPCs in various anatomical sites were significantly increased during follow-up. Particularly, the risk is significantly high among thyroid cancer survivors aged < 40 years and within 5 years of diagnosis. Medical providers should consider the long-term follow-up of patients with thyroid cancer and discuss the risk of SPC, especially if they complain of cancer-related symptoms.

## Figures and Tables

**Figure 1 diagnostics-13-02903-f001:**
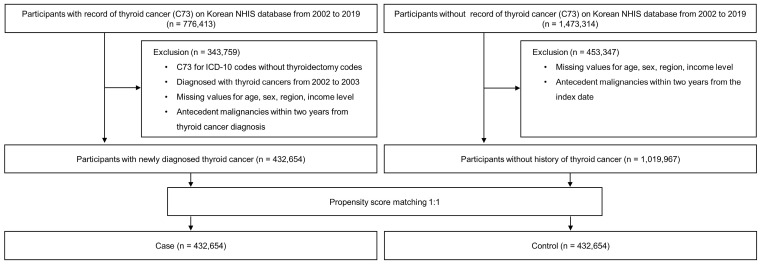
Flow diagram of study cohort selection.

**Figure 2 diagnostics-13-02903-f002:**
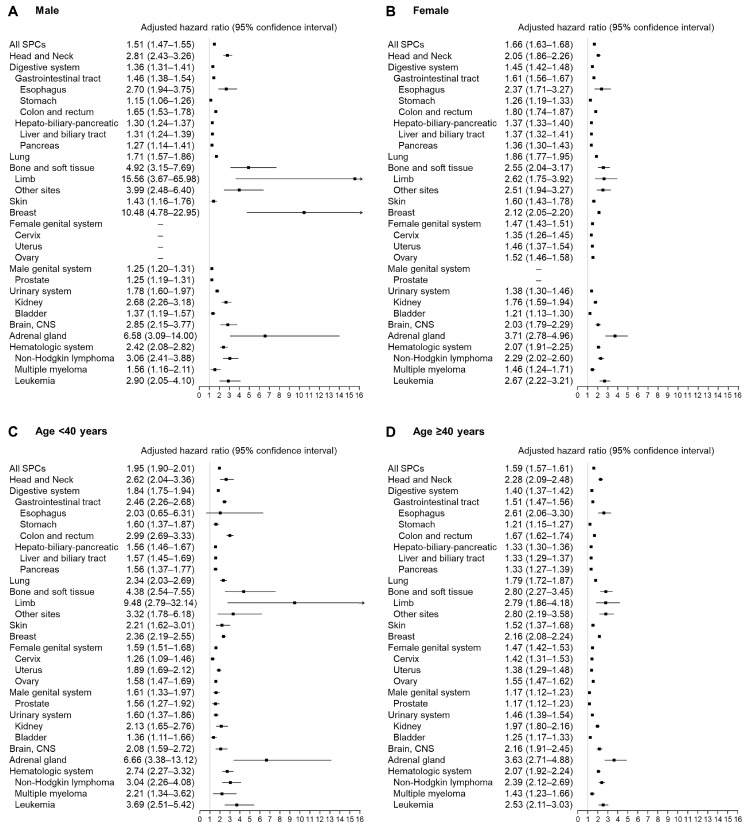
Adjusted hazard ratios of subsequent primary cancer in patients with thyroid cancer compared with matched controls according to sex ((**A**) male; (**B**) female) and age group ((**C**) <40 years; (**D**) ≥40 years).

**Figure 3 diagnostics-13-02903-f003:**
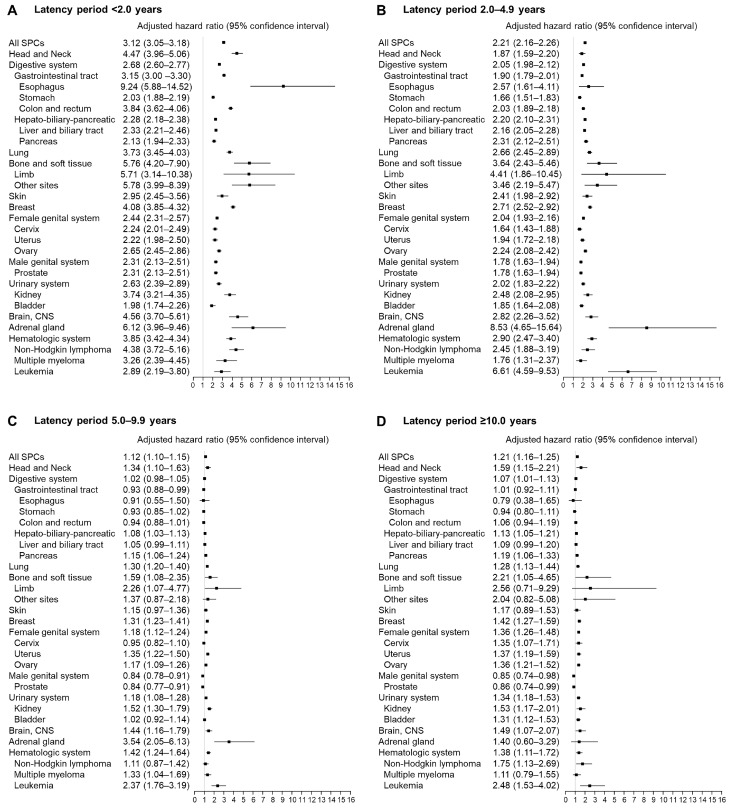
Adjusted hazard ratios of subsequent primary cancer in patients with thyroid cancer compared with matched controls according to latency period ((**A**) <2.0 years; (**B**) 2.0–4.9 years; (**C**) 5.0–9.9 years; and (**D**) ≥10.0 years).

**Table 1 diagnostics-13-02903-t001:** Baseline characteristics of study participants.

	Overall	Case	Control	*p*-Value
Total number, *n*	865,308	432,654	432,654	
Age (years), mean ± SD	48.6 ± 12.2	48.6 ± 12.2	48.6 ± 12.2	0.930
Sex, *n* (%)				1.000
Male	164,939 (19.1)	82,470 (19.1)	82,469 (19.1)	
Female	700,369 (80.9)	350,184 (80.9)	350,185 (80.9)	
Region of residence, *n* (%)				0.890
Urban	422,549 (48.8)	211,307 (48.8)	211,242 (48.8)	
Rural	442,759 (51.2)	221,347 (51.2)	221,412 (51.2)	
Income level, *n* (%)				1.000
1 (lowest)	133,490 (15.5)	66,745 (15.4)	66,745 (15.4)	
2	114,044 (13.2)	57,022 (13.2)	57,022 (13.2)	
3	140,750 (16.3)	70,375 (16.3)	70,375 (16.3)	
4	193,973 (22.4)	96,986 (22.4)	96,987 (22.4)	
5 (highest)	283,051 (32.7)	141,526 (32.7)	141,525 (32.7)	
CCI score, *n* (%)				<0.001
0	314,134 (51.8)	119,726 (27.7)	194,408 (44.9)	
1	112,833 (13.0)	56,715 (13.1)	56,118 (13.0)	
2	73,239 (8.5)	38,655 (8.9)	34,584 (8.0)	
3	53,871 (6.2)	30,388 (7.0)	23,483 (5.4)	
≥4	311,231 (36.0)	187,170 (43.3)	124,061 (28.7)	
Subgroup ^a^				
n (%)	258,585	141,952	116,633	
Obesity ^b^				<0.001
Underweight	7324 (2.8)	3472 (2.4)	3852 (3.3)	
Normal	100,905 (39.0)	52,170 (36.8)	48,735 (41.8)	
Overweight	63,497 (24.6)	34,679 (24.4)	28,818 (24.7)	
Obesity I	75,443 (29.2)	44,316 (31.2)	31,127 (26.7)	
Obesity II	11,416 (4.4)	7315 (5.2)	4101 (3.5)	
Smoking				<0.001
Never	211,094 (81.6)	116,839 (82.3)	94,255 (80.8)	
Former	22,508 (8.7)	13,408 (9.4)	9100 (7.8)	
Current	24,983 (9.7)	11,705 (8.2)	13,278 (11.4)	

CCI, Charlson Comorbidity index. ^a^ Subgroup of participants with data on body mass index and smoking status retrieved within 1 year from index date. ^b^ Body mass index (kg/m^2^) was categorized as <18.5 (underweight), ≥18.5 to <23 (normal), ≥23 to <25 (overweight), ≥25 to <30 (obesity I), and ≥30 (obesity II).

**Table 2 diagnostics-13-02903-t002:** Incidence and hazard ratios of subsequent primary cancer for patients with thyroid cancer and matched controls.

Type of SPC	Case (*n* = 432,654)	Control (*n* = 432,654)	Crude HR (95% CI)	Adjusted HR ^a^ (95% CI)
All SPCs	78,584 (18.2)	49,979 (11.6)	1.84 (1.82–1.86)	1.62 (1.60–1.64)
Head and Neck (C00–C14, C30–C32)	2070 (0.5)	853 (0.2)	2.68 (2.47–2.90)	2.27 (2.10–2.47)
Digestive system (C15–C26)	31,518 (7.3)	22,066 (5.1)	1.67 (1.64–1.64)	1.43 (1.41–1.46)
Gastrointestinal tract (C15–C20)	14,204 (3.3)	8889 (2.1)	1.83 (1.78–1.88)	1.57 (1.53–1.62)
Esophagus (C15)	283 (0.1)	103 (0.0)	3.09 (2.47–3.88)	2.53 (2.01–3.18)
Stomach (C16)	4095 (1.0)	3258 (0.8)	1.44 (1.38–1.51)	1.22 (1.17–1.28)
Colon and rectum (C18–C20)	9686 (2.2)	5429(1.3)	2.04 (1.97–2.11)	1.77 (1.71–1.83)
Hepato-biliary-pancreatic (C22–C25)	16,548 (3.8)	12,432 (2.9)	1.58 (1.54–1.61)	1.35 (1.32–1.38)
Liver and biliary tract (C22–C24)	11,483 (2.7)	8618 (1.9)	1.56 (1.51–1.60)	1.35 (1.31–1.39)
Pancreas (C25)	5065 (1.2)	3814 (0.9)	1.62 (1.55–1.69)	1.35 (1.29–1.41)
Lung (C34)	6613 (1.5)	3682 (0.9)	2.12 (2.04–2.21)	1.82 (1.74–1.89)
Bone and soft tissue (C40–C41)	420 (0.1)	138 (0.0)	3.40 (2.81–4.13)	2.96 (2.43–3.59)
Limb (C40)	118 (0.03)	36 (0.0)	3.76 (2.58–5.47)	3.32 (2.27–4.86)
Other sites (C41)	302 (0.07)	102 (0.0)	3.28 (2.61–4.11)	2.83 (2.25–3.55)
Skin (C43–44)	1096 (0.3)	731 (0.2)	1.79 (1.63–1.97)	1.56 (1.42–1.72)
Breast (C50)	10,458 (2.4)	5173 (1.2)	2.32 (2.24–2.40)	2.16 (2.09–2.24)
Female genital system (C51–C58)	10,810 (2.5)	8312 (1.9)	1.55 (1.51–1.60)	1.50 (1.45–1.54)
Cervix (C53)	1975 (0.5)	1518 (0.4)	1.48 (1.39–1.59)	1.37 (1.28–1.47)
Uterus (C54–C55)	2689 (0.6)	2123 (0.5)	1.56 (1.48–1.65)	1.49 (1.40–1.58)
Ovary (C56–C57)	6090 (1.4)	4615 (1.1)	1.57 (1.51–1.63)	1.55 (1.49–1.61)
Male genital system (C60–C63)	4549 (1.1)	3815 (0.9)	1.40 (1.40–1.46)	1.18 (1.13–1.23)
Prostate (C61)	4478 (1.0)	3764 (0.9)	1.40 (1.37–1.45)	1.17 (1.12–1.23)
Urinary system (C64–C68)	4177 (1.0)	2940 (0.7)	1.71 (1.63–1.79)	1.46 (1.39–1.54)
Kidney (C64)	1633 (0.4)	821 (0.2)	2.31 (2.12–2.51)	1.97 (1.80–2.14)
Bladder (C67)	2292 (0.5)	1940 (0.5)	1.45 (1.63–1.54)	1.25 (1.17–1.33)
Eye, orbit (C69)	73 (0.0)	41 (0.0)	2.14 (1.46–3.16)	1.96 (1.32–2.91)
Brain, CNS (C70–C72)	968 (0.2)	468 (0.1)	2.41 (2.16–2.70)	2.15 (1.92–2.40)
Adrenal gland (C74)	278 (0.1)	68 (0.0)	4.63 (3.55–6.04)	4.07 (3.11–5.32)
Hematologic system (C81–C96)	2479 (0.6)	1135 (0.3)	2.52 (2.35–2.71)	2.15 (2.00–2.31)
Hodgkin Disease (C81)	54 (0.0)	20 (0.0)	3.05 (1.82–5.11)	2.79 (1.65–4.70)
Non-Hodgkin lymphoma (C82–C86)	1165 (0.3)	440 (0.1)	2.93 (2.62–3.27)	2.45 (2.19–2.74)
Multiple myeloma (C90)	494 (0.1)	340 (0.1)	1.80 (1.57–2.07)	1.48 (1.28–1.70)
Leukemia (C91–C95)	547 (0.1)	209 (0.1)	3.07 (2.61–3.60)	2.72 (2.31–3.20)

SPC, subsequent primary cancer; HR, hazard ratio; CI, confidence interval; C, diagnostic code used in the International Classification of Diseases (ICD)-10; and CNS, central nervous system. Hazard ratios were determined using Cox proportional hazard models. ^a^ Adjusted HR for Charlson Comorbidity index.

## Data Availability

The researcher is not legally permitted to release the data. The entire dataset is accessible through the NHIS database, which can be accessed at https://nhiss.nhis.or.kr/. NHIS service provides access to this data to any researcher who commits to adhering to research ethics, albeit for a fee. If you want to access the data of this article, you can download it from the website following an assurance of compliance with research ethics.

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
