# Peer review of "Risk of Subsequent Primary Cancer in Thyroid Cancer Survivors: A Nationwide Population-Based Study"

_diagnostics, 2023, doi:10.3390/diagnostics13182903_

Round 1

Reviewer 1 Report

Thank you for inviting me to review this large nationwide study that looks at the relationship between secondary primary cancer in patients with thyroid cancer. This study is a retrospective controlled study looking at the association between secondary cancers in patients after receiving treatment for thyroid cancer. Although this study examines a large cohort of patients and is adequately powered, it does not mention the stage of thyroid cancer or dose of RAI treatment or whether external beam radiation was used as an adjunct to surgery. This factor is important to consider since such treatment is a known risk factor for developing a secondary malignancy. Without this information it is difficult to conclude a causal relationship between the two. I am concerned that some important findings are being lost without these details. Additionally, the extent of surgery that is performed and the dose of RAI have all changed in the last decade which may impact the outcomes and findings of this large cohort study. A subgroup analysis would be useful to determine what treatment was received if any (the dose of RAI, whether external beam radiation was administered) and the stage of the thyroid cancer if possible.  

Was the cancer of the thyroid all papillary thyroid cancer or did this include the other subtypes such as follicular, medullary and anaplastic thyroid cancer? Again, this is an important distinction since each subtype behaves differently. I note that at the end of the paper the author's mention they were not able to gather information on the thyroid cancer subtype. 

Author Response

We agree with the reviewer’s comments and appreciate the valuable comments. Unfortunately, however, it is practically difficult to conduct a subgroup analysis according to the cumulative dose of RAI and EBRT. For accessing the nationwide Korean NHIS database, it is a system that can only be analyzed within the assigned room after making a reservation at the data analysis center, and there are several restrictions. Therefore, given the current situation, it would not be able to analyze what the reviewer stated in a short period of time. In addition, as we mentioned as a limitation, there is no information on the cancer stage and subtypes of thyroid cancer in the NHIS database (only claim codes for thyroid cancer [C73] provided).

Therfore, we have included these points as limitations in the Discussion section [Page 12; Lines 240-248; Discussion section].

Nevertheless, our study had some limitations. First, although we confirmed the association between thyroid cancer and SPCs, proving the causality was difficult because multiple factors are involved in carcinogenesis. Thus, we tried to control the confounding factors using propensity score matching, multivariate adjusted models, and subgroup analyses. However, the effect of treatment, such as the dose of RAI and external beam radiation, was not considered, which is a limitation. Further studies on the causal relationship and mechanisms are needed. Second, due to limitations of the NHIS database, the cancer stages, and histologic subtypes of thyroid cancer, such as papillary, follicular, medullary, and anaplastic subtypes, were not available.

Reviewer 2 Report

This is an interesting large cohort study that investigated the risk of subsequent primary cancer in thyroid cancer survivors in the whole Korean population. I need some explanations from the authors regarding the following points:

1. I don't understand the purpose of excluding patients who who had not undergone thyroidectomy and those diagnosed with thyroid cancer from 2002 to 2003? The authors mentioned that this was done to ensure the inclusion of patients newly diagnosed with thyroid cancer, which seems to me irrelevant. Why would patients diagnosed in 2004 considered "newly diagnosed" while those diagnosed only one year earlier considered "not newly diagnosed"?????!!!!!!!

2. Line 84: What is meant by the following numbers (P4551, P4552, P4553, P4554, and P4561)?

3. It is mentioned in the statistical analysis that all continuous variables were presented as means and SD and compared by t-test. This assumes that all those data were parametric. However, it is not mentioned at all that the normality of data was tested.

The language of the manuscript is acceptable. Further moderate revision is recommended.

Author Response

  1. I don't understand the purpose of excluding patients who had not undergone thyroidectomy and those diagnosed with thyroid cancer from 2002 to 2003? The authors mentioned that this was done to ensure the inclusion of patients newly diagnosed with thyroid cancer, which seems to me irrelevant. Why would patients have diagnosed in 2004 considered "newly diagnosed" while those diagnosed only one year earlier considered "not newly diagnosed"?????!!!!!!!

Response 1:

We used data from the period between 2002 and 2019, excluding the initial 2 years (2002 and 2003) as a washout period. For example, a patient diagnosed with thyroid cancer in 2001 would still have a thyroid cancer claim code (C73) in 2002. Although this patient had developed thyroid cancer in 2001, he/she is counted as an incident case in our cohort for the year 2002. Considering the follow-up period for thyroid cancer patients in clinical practice, during the initial 2 years (2002 and 2003), patients diagnosed prior to 2002 could be mixed in, potentially leading to an overestimation of incidence. To address this, we applied a 2-year washout period and considered patients diagnosed with thyroid cancer from 2004 onwards as newly diagnosed cases.

We mentioned the reason for excluding the first 2 years [Page 2; Lines 85-86; Materials & Methods section].

The initial 2 years (2002 and 2003) were excluded as a washout period due to the possibility of including patients diagnosed with thyroid cancer prior to the study period.

  1. Line 84: What is meant by the following numbers (P4551, P4552, P4553, P4554, and P4561)?

Response 2:

They are claim codes of thyroid surgery: P4551 [Total Thyroidectomy-Unilateral], P4552 [Total Thyroidectomy-Bilateral], P4553 [Subtotal Thyroidectomy-Unilateral], P4554 [Subtotal Thyroidectomy-Bilateral], and P4561 [Radical Operation of Malignant Thyroid Tumor].

We added ‘claim codes’ to that part for a better understanding [Page 2; Line 84; Materials & Methods section].

To ensure the inclusion of patients newly diagnosed with thyroid cancer, we excluded patients who had not undergone thyroidectomy (claim codes: P4551, P4552, P4553, P4554, and P4561) and those diagnosed with thyroid cancer from 2002 to 2003.

  1. It is mentioned in the statistical analysis that all continuous variables were presented as means and SD and compared by t-test. This assumes that all those data were parametric. However, it is not mentioned at all that the normality of data was tested.

Response 3:

We compared two sufficiently large samples from a population (case vs. control), with a total of 865,308 individuals, 432,654 in each group. Therefore, in accordance with the central limit theorem, which posits that the distribution of a sample variable approximates a normal distribution as the sample size increases, the variable was represented using the mean and standard deviation, and a t-test was employed for comparison.